# The effect of an mHealth application based on continuous support and education on fear of childbirth, self-efficacy, and birth mode in primiparous women: A randomized controlled trial

**Sahar Khademioore**[1¤]*, **Elham Ebrahimi**[1], **Ahmad Khosravi**[2], **Shohreh Movahedi**[3]

**1** Department of Reproductive Health, School of Nursing and Midwifery, Tehran University of Medical Sciences, Tehran, Iran, **2** Department of Epidemiology, Center for Health Related Social and Behavioral Sciences Research, Shahroud University of Medical Sciences, Shahroud, Iran, **3** Department of Obstetrics and Gynecology Tehran University of Medical Sciences, Tehran, Iran

¤ Current address: Department of Health Research Methods, Evidence & Impact, McMaster University, Hamilton, Ontario, Canada

* khades1@mcmaster.ca

**Data Availability Statement:** The authors confirm all the data collected in this study has been thoroughly presented within the paper. However,

## Abstract

### Background

The Fear of Childbirth (FOC) is associated with several adverse health outcomes for children and women. This study aimed to evaluate the effectiveness of an interactive mobile health application named Tele-midwifery with an emphasis on continuous care and education, on FOC, self-efficacy, and childbirth mode in primiparous women.

### Methods

Seventy primiparous women attending the prenatal clinic of Baharlou Hospital in Tehran, Iran, were randomly assigned to two parallel intervention and control groups with 35 participants each. Women in the intervention group received Tele-midwifery for eight weeks, whereas women in the control group only received routine care. The Wijma delivery expectancy/experience questionnaire and the Childbirth Self-Efficacy Inventory were used to measure the FOC and self-efficacy at baseline and eight weeks after the intervention. The FOC and birth mode were also measured after birth.

### Results

There was a significant decrease in FOC among women in the intervention group compared to control groups after eight weeks of intervention (- 20.9 [95% Confidence Interval,—24.01 to—17.83], p < 0.001), and after birth (- 30.8, [95% CI—33.8 to—27.97], p < 0.001). After eight weeks, the mean self-efficacy score in the intervention group was significantly higher than the control group (p < 0.001). Compared to the control group, the intervention group had a lower C-Section (CS) rate (p = 0.03).

this study was conducted in a single center over a short period of time, so making the data set public may lead to participant identification. Data set will be available upon request from the ethics committee of Tehran University of Medical Sciences at Ethics@sina.tums.ac.ir.

**Funding:** The authors received no specific funding for this work.

**Competing interests:** The authors have declared that no competing interests exist.

## Conclusions

Tele-midwifery intervention reduced FOC, increased women's self-efficacy in childbirth, and decreased the number of CS in a group of first-time mothers. Healthcare providers can use the mHealth approach to support pregnant women with FOC.

## Trial registration

Registration number: IRCT20200122046227N1, Registered on 27 January 2020.

## Introduction

Fear of childbirth (FOC) is a well-known problem affecting women, especially primiparous during pregnancy and postpartum period [1]. FOC is defined as severe anxiety related to childbirth, which can manifest through sleep disorders, physical complaints, and lack of focus on work or social life [2]. According to a large-scale systematic review and meta-analysis of 29 studies on 853,988 pregnancies, the global prevalence of FOC was reported to be 14 percent (95% Confidence Interval (CI) 0.12–0.16), with an increasing trend in recent years [3]. Women suffering from FOC are at increased risk of maternal and child health complications such as prolonged labor, increased use of analgesia, higher rate of emergency and elective and C-Section (CS), post-traumatic stress disorder, and postpartum depression [4–6]. Previous studies showed that FOC is also strongly associated with a lack of childbirth self-efficacy, especially in first-time mothers [7]. Self-efficacy plays a vital role in adopting to pregnancy and childbirth, and can affect women's attitude and motivation for a vaginal birth rather than requesting CS [8]; As a result, the adverse effects of CS on mothers and their children will be prevented, and the unnecessary costs for the healthcare system due to the high rate of CS will be reduced [9].

Even though various interventions such as yoga, art therapy, and counseling, have been shown to be effective in reducing FOC, there is no consensus regarding the best treatment for alleviating women's fears [10]. Furthermore, some of these interventions are time-consuming, expensive, dissatisfying to pregnant women, and difficult to implement [11]. Due to these challenges, technology-based methods, such as Mobile health applications (mHealth), have gained attention in various fields to improve women's health (e.g., mental health) [12]. A key advantage of mHealth is the accessibility of healthcare services, such as educational content or consultations, regardless of time or space constraints [13]. Pregnant women report greater satisfaction in receiving pregnancy-related services through mobile applications [14, 15], which offer continuous care and lead to a positive pregnancy and childbirth experience [16]. mHealth is an efficient and popular technology that has women's acceptance and can improve one's ability to face various challenging situations [17]. Providing education and continuous support for pregnant women can eliminate many causes of FOC and improve their self-efficacy, contributing to better preparedness for pregnancy and childbirth [18, 19]. Receiving education via mHealth applications and providing continuous care can lead to greater sense of responsibility for obtaining information and learning, which results in in-depth knowledge and awareness [20].

Preventing FOC and promoting positive attitudes towards childbirth during pregnancy are critical factors in reducing the risk of traumatic childbirth and other complications associated with pregnancy and childbirth. Currently, there is no systematic approach for screening and treating women with FOC [21]. Considering the lack of specialized services for

women with FOC, integrating effective interventions such as an mHealth application based on education and continuous support with the healthcare system can be an effective approach for providing tailored care to pregnant women. The application of mHealth based on the education and continuous support on improving the FOC, has received little attention, especially among Iranian pregnant women. Thus, in this study, an mHealth application was considered a possible strategy for treating women with FOC, improving their self-efficacy, and reducing the CS rate.

## Materials and methods

### Study design and eligibility criteria

We conducted a Randomized Clinical Trial (RCT) to evaluate the effectiveness of using an mHealth application called Tele-midwifery to reduce FOC, increase childbirth self-efficacy, and reduce the number of CS among primiparous women. The research setting was the perinatal clinic of Baharlou Hospital in Tehran, Iran, and enrollment took place from February to April 2020.

Inclusion criteria in this study were: being primiparous, having FOC (confirmed by the score of 38 and above on the Wijma delivery expectancy/experience questionnaire (W-DEQ)), being between 18 to 40 years of age, pregnant with a singleton fetus, 26–29 weeks of pregnancy, having access to a smart device such as a smartphone, or a tablet (women or their spouses), as well as the ability to work with them, having access to the Internet, not having chronic diseases and CS indications before or during pregnancy. The exclusion criteria were, emergency pregnancy conditions that required intervention (e.g., severe symptoms of COVID-19, placental abruption, fetus abnormalities, preeclampsia).

### Outcomes

The primary outcome of this study was a change in FOC score. The secondary outcomes were a change in childbirth self-efficacy score, and childbirth mode.

### Measures

In this study, the fear related to childbirth in the prenatal and postnatal period was measured using W-DEQ versions A and B. W-DEQ version A evaluates women's prenatal expectances before childbirth, and W-DEQ version B evaluates experiences with recent childbirth. Each version contains 33 items with a 6-point Likert scale ranging from 0 (extremely) to 5 (not at all), and scores range from 0 to 165, with higher scores indicating higher fear of childbirth [22]. According to this questionnaire, a score less than or equal to 37 represent mild fear, a score of 38–65 represents a moderate level, 66–84 represents high level of fear, and a score more than 85 shows a severe level of fear. The reliability of the W-DEQ version A and B questionnaire for primiparous women was confirmed with Cronbach's alpha of 0.89 and 0.92, respectively [22]. The validity and reliability of the Persian version of this questionnaire for Iranian women have been confirmed [23].

The Childbirth Self-Efficacy Inventory (CBSEI) is a self-report instrument that measures outcome expectancies (OE) and efficacy expectancies (EE) for coping with an approaching childbirth experience. This questionnaire has two parts and 62 items that scored on a ten-point Likert scale (1 = not at all; 10 = completely sure), with a higher score indicating greater childbirth self-efficacy. The reliability of the questionnaire has been confirmed with Cronbach's alpha 0.86–0.95 [24]. The validity and reliability of the Persian version of this

questionnaire were assessed by Khorsandi et al. (2008). In their study, high internal consistency with Cronbach's alpha coefficient of 0.84 to 0.91 was reported for the CBSEI [25].

## Sample size

The sample size was calculated based on our primary outcome which was FOC. We referred to the results of Gözde Birsbir et al.'s (2016) study on the effect of antenatal education on FOC to estimate our sample size [26]. A two-sided t-test of difference between means with a power of 90%, the type one error equal to 0.05, a mean difference of 21 scores between groups on the W-DEQ scale for FOC was considered to estimate the required number of participants. In the sample size calculation, we also accounted for a 20% attrition rate. A sample size of 70 pregnant women (35 in each group) was estimated.

## Procedures

A convenience sampling approach was used, and a consecutive sample of pregnant women attending to the selected prenatal clinic for receiving their routine prenatal care were evaluated using their medical records for inclusion and exclusion criteria, by a colleague not involved in the study. Afterward, the objectives of the study were explained to eligible pregnant women. After consent was obtained, they were asked to complete the W-DEQ version A. women with a score of 38 and above (moderate and high FOC score) were enrolled in the study. A total of 214 pregnant women were evaluated for recruitment in the study. Specifically, 98 women did not meet the inclusion criteria, and 32 women declined to participate. Overall, 70 eligible individuals were randomly assigned to intervention (Tele-midwifery application) and control groups, with 35 participants in each (Fig 1).

A randomized allocation into two groups of intervention and control with a 1: 1 ratio took place during the routine prenatal visit in the hospital. A colleague who was not involved in the study put each computerized generated random sequences separately in sealed and opaque envelopes. To mask the person performing the allocation, the letters A and B were used to assign women to the intervention or control group. In both intervention and control groups, the Tele-midwifery application was installed on their smartphones during their prenatal visit, and instructions for using the application were explained. This application only allowed access to the questionnaires in the post-test, and follow-up stages for participants in the control group. However, the intervention group had access to all features of the application. The researcher could track the women's activities in the application, and in case the women had not use the application for more than two days, a reminder was sent to follow them up. To prevent information contamination in the control group, after installing the application and before activating their account, women selected their group type as A or B. In addition, we asked the women in both groups to avoid sharing the contents of the application with each other for the duration of the trial. The Tele-midwifery application had two main stages for answering the questions. After filling out the baseline questionnaire in the initial registration, including demographic, W-DEQ, and CBSE questionnaires, each groups of participants completed the W-DEQ and CBSE questionnaires again at 34–36 weeks (after eight weeks of receiving the intervention). Participants were also asked to inform the researcher by sending a message after hospitalization for their birth, whether in the Tele-midwifery application or by contacting the researcher directly by her phone number. Afterward, the W-DEQ version B questionnaire appeared in the application within the first two hours after birth, and the birth mode was recorded. Due to the nature of the intervention, it was not possible to mask participants from the group allocation. However, investigators, care providers, data collectors, and statisticians were masked from the allocation.

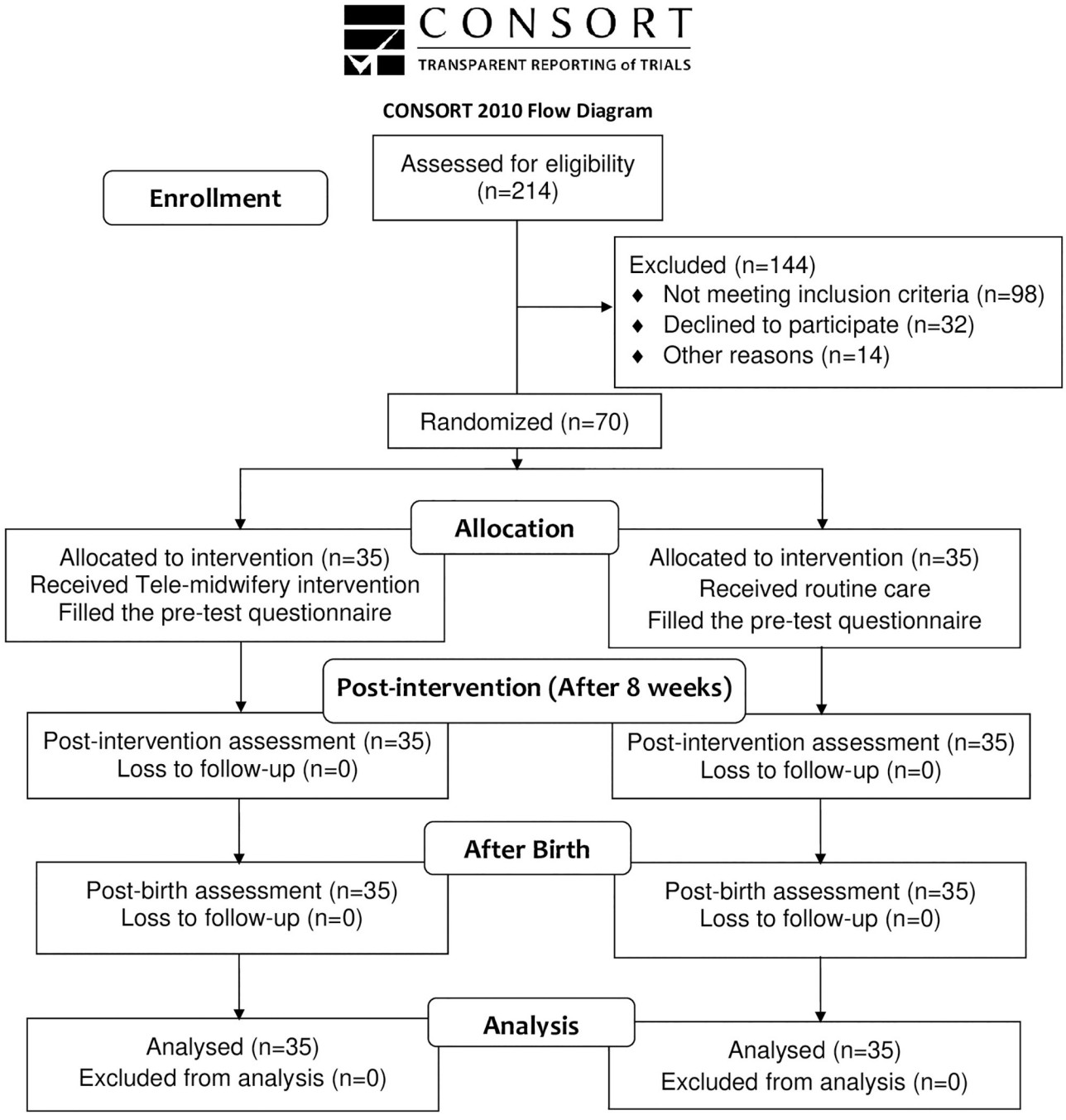

**Fig 1. CONSORT flow diagram of the study.**

## Tele-midwifery application

First, to prepare the Tele-midwifery application's content, reliable and up-to-date scientific sources such as textbooks, scientific articles, related guidelines, studied carefully by the research team, and necessary content in various fields were reviewed [10, 27–30]. This content included videos, audios and short texts with related images produced by the research team to

make it understandable for all women with any level of education and increase attractiveness. In the next stage, the content's validity was evaluated by three relevant experts (reproductive health, gynecologist, and mental health specialist), then sent to software experts in a format that can be used as a mobile application. Finally, the application was used as a trial version by five pregnant women who were not involved in the study, in which aspects such as comprehensibility, simplicity of its contents, and user-friendliness were checked. Then the comments of these users were applied to modify the application, and the final version was prepared.

## Dimensions of intervention

**Education.** The educational content of the Tele-midwifery application was designed to include all possible causes of FOC in Iranian women, such as fear of labor pain, lack of information related to the mother and baby's health, what women should expect during labor and birth, the characteristics of each stage of labor, misconceptions such as distrust of healthcare providers, and complications of vaginal birth [31, 32]. The list of educational content is shown in Table 1. The educational content included information and exercises to increase women's knowledge and challenge their underlying beliefs causing the FOC. Besides the educational information in the form of text, audio, video, and image, this dynamic application provides the opportunity for women receiving the intervention to ask their question regarding the educational content within a chat box available in the application. Educational content was also accessible offline for women on an archive in the application. The application's educational content was designed for eight weeks in a way that between 3–4 short messages were sent to women on a daily basis. After eight weeks, women in both the intervention and control groups received the next stage (post-test) questionnaires.

**Table 1. Five components of educational content of Tele-midwifery application.**

| Content | Details | Teaching methods and number |
|---------|---------|------------------------------|
| 1. Pregnancy | The third trimester complications, healthy eating, relaxation technique, emotional change in pregnancy, fetal development, female anatomy, sharing feelings and concerns related to pregnancy, COVID-19 symptoms, and preventive measures | Videos: 2<br>Texts: 23<br>Pictures: 10<br>Audio: 2 |
| 2. Labor and childbirth | Signs of labor, different stages of labor, coping techniques with labor, including, breathing and relaxation exercises, birth positions, massage, hydrotherapy, aromatherapy, pushing techniques, birth support partner, common myth about labor contractions, share feelings and thoughts about labor and childbirth, identify feeling about childbirth, visualization of childbirth | Video: 1 positive childbirth experience stories video and audio: 3<br>pictures:5<br>Texts: 10 |
| 3. Interventions during the labor and childbirth | Induction and augmentation of labor, episiotomy, epidural, amniotomy, vaginal examinations in labor, preparing a birth plan, visualization of childbirth according to the birth plan | Videos: 1<br>Texts: 10<br>Pictures: 5<br>Audio: 3 |
| 4. Childbirth mode | Cesarean section: indications, side effects, surgery preparation, recovery, operative vaginal birth: forceps or a vacuum extractor | Videos: 1<br>Texts: 5<br>Pictures: 2<br>Audio:1 |
| 5. Postpartum | Physical and psychological changes during the postpartum period, parenthood, mother-baby interaction, breastfeeding, newborn care, nutritional care, postpartum sexuality, family planning, sharing feeling and thought about the postpartum period | Videos: 1<br>Texts: 15<br>Pictures: 8<br>Audio: 4 |

**Continuous support and engagement of women.** To provide continuity of care and support between two prenatal visits, participants in the intervention group, could contact the researchers if they had concerns and questions about their pregnancy and birth care. The team of researchers including midwives and obstetricians were available throughout the day (between 7 a.m. to 9 p.m.), to provide women with an accessible source of information about their pregnancy and childbirth decisions. However, this feature was not intended to substitute the usual antenatal care, and no medication prescriptions, requesting laboratory tests, or imaging were offered via our mHealth. They could also write about their feelings and experiences publicly in the forum linked to the educational contents, and pregnant women had the opportunity to exchange their feelings with each other under the research team's supervision. The mHealth application can also help women in expressing their feelings and symptoms due to the anonymity of this method of delivering the intervention [33].

The control group received only routine prenatal care. Currently there are no specific guidelines in Iran, for providing care to pregnant women suffering from FOC. The routine prenatal care in Iran includes up to 8 prenatal visits, which in the study site would be provided to all pregnant individuals by midwives and obstetricians. Routine prenatal care includes regular visits to monitor women and fetal health. To better access women in the control group and limit in-person interaction due to the COVID-19 pandemic, we asked them not to uninstall the application from their mobile phones until birth and answer questionnaires after eight weeks and after childbirth.

## Statistical analyzes

The data analyzed by IBM SPSS statistics version 25 and p-value > 0.05 was considered as a statistically significant level. To compare statistical differences of demographic data in intervention and control groups, an independent t-test, Chi-square test, and Fisher's exact test were used. The repeated-measures analysis of variance (ANOVA) was used to compare the mean score of FOC between intervention and control groups at different time points (baseline, after 8 weeks, after childbirth). Linear regression models used to estimate the change in FOC and self-efficacy score in the intervention group compared to the control group in different time points. Also, for analyzing the birth mode, Chi-square were used to determine the differences between the groups. A Logistic regression model was also performed to estimate the odds of CS in the intervention group compared to the control group.

## Ethical considerations

The ethics committee of Tehran University of Medical Sciences approved the research procedure (IR.TUMS.FNM.REC.1398.135). At the beginning of the study, the researchers provided the necessary explanations about the intervention and the study's objectives to the participant. After receiving sufficient information from various aspects of the research, all participants provided a signed written informed consent, and they could leave at any stage of the research. Each woman was assigned a unique code to ensure the confidentiality of their medical information, which prevented the identification of the participants.

## Results

A total of 70 nulliparous women with FOC enrolled in the study in two intervention (n = 35) and control (n = 35) groups. There was no loss to follow-up, and no exclusion from the study based on our criteria, and although we prepared to refer participants in case of an emergency, no such contacts were made during the study period and all 70 participants were included in the final analysis. In this study, the mean age of participants in the intervention group was 24.3

**Table 2. Characteristics of the participants.** (n = 70).

| Characteristics | Intervention (n = 35) | Control (n = 35) | P-value |
|---|---|---|---|
| **Women's age (Mean ±SD, years)** | 24.3 ± 3.5 | 25.6 ± 3.5 | 0.1*** |
| **Spouse age (Mean ±SD, years)** | 28.8 ± 3.4 | 29.5 ± 3.1 | 0.3*** |
| **Gestational age (Mean ±SD, weeks)** | 27.7 ± 0.8 | 27.4 ± 0.7 | 0.1*** |
| **Women's education status n (%)** | | | |
| Primary/secondary school | 4 (11.4) | 4 (11.4) | 0.7* |
| High school/diploma | 23 (65.7) | 20 (57.1) | |
| University | 8 (22.9) | 11 (31.5) | |
| **Spouse education status n (%)** | | | |
| Primary/secondary school | 8 (22.8) | 6 (17.1) | 0.3* |
| High school/diploma | 19 (54.3) | 22 (62.9) | |
| University | 8 (22.9) | 7 (20.0) | |
| **Family income n (%)** | | | |
| Less than expenses | 12 (34.3) | 9 (25.7) | 0.5* |
| Equal to expenses | 20 (57.1) | 20 (57.2) | |
| More than expenses | 3 (8.6) | 6 (17.1) | |
| **Women occupation n (%)** | | | |
| Housewife | 19 (54.3) | 22 (62.9) | 0.4** |
| Employee | 16 (45.7) | 13 (37.1) | |
| **Spouse occupation n (%)** | | | |
| Laborer | 6 (17.1) | 2 (5.7) | 0.3* |
| Governmental | 11 (31.4) | 11 (31.4) | |
| Nongovernmental | 18 (51.5) | 22 (62.9) | |
| **Pregnancy intention n (%)** | | | |
| Intended | 24 (68.6) | 19 (54.3) | 0.2** |
| Unintended | 11 (31.4) | 16 (45.7) | |

*Fisher Exact Test

**Chi square test

***Independent t-test

years, and 25.6 years in the control group. The majority of women's gestational age in the intervention and control groups at the beginning of the study was 28 weeks (40% in the intervention group and 45.7% in the control group). According to Table 2, there was no statistically significant difference between the groups regarding sociodemographic characteristics (p > 0.05).

## FOC

A repeated-measures ANOVA with a Greenhouse-Geisser correction for within-subject effects of FOC scores showed that time and the group interaction effect is significant for this variable (time*group: F = 131, p < 0.001). To compare groups at different time-points (baseline, after 8 weeks, and after childbirth), linear regression model was used. There was no statistically significant difference in mean scores of FOC in the intervention and control groups at the baseline (p = 0.10). The result of linear regression model showed that on average the estimated score of FOC for individual in the intervention group is 21.14 scores lower than those in the control group (95% CI [-23.94, -18.34]; p <0.0001, $R^2$ = 0.77) after 8 weeks of intervention. Furthermore, the FOC score after childbirth was improved by an estimated of 31.23 score more on

**Table 3. Effect of mHealth application on FOC score in intervention group compared to control group at different time points (n = 70).**

| Time/ Group | Coefficient | SE | 95% CI | | P-Value |
|---|---|---|---|---|---|
| | | | LL | UL | |
| **Baseline** | | | | | |
| Intervention vs. Control | 2.83 | 1.70 | -0.56 | 6.21 | 0.10 |
| **After 8 weeks** | | | | | |
| Intervention vs. Control | - 21.14 | 1.40 | -23.94 | -18.34 | <0.0001 |
| **After childbirth** | | | | | |
| Intervention vs. Control | -31.23 | 1.30 | -33.82 | -28.64 | <0.0001 |

SE = Standard Error, CI = Confidence Interval, LL = lower Limit, UL = Upper Limit

average in the intervention group compared to those in the control group (95% CI [-33.82, -28.64], p <0.0001, $R^2$ = 0 .90) (Table 3). The findings remained consistent after adjusting for age, income, education, and baseline fear (S1 Table in S3 File). In summary, the result shows that on average the intervention group had statistically significantly lower FOC scores compared to the control group after 8 weeks and after birth (Fig 2, S2 Table in S3 File).

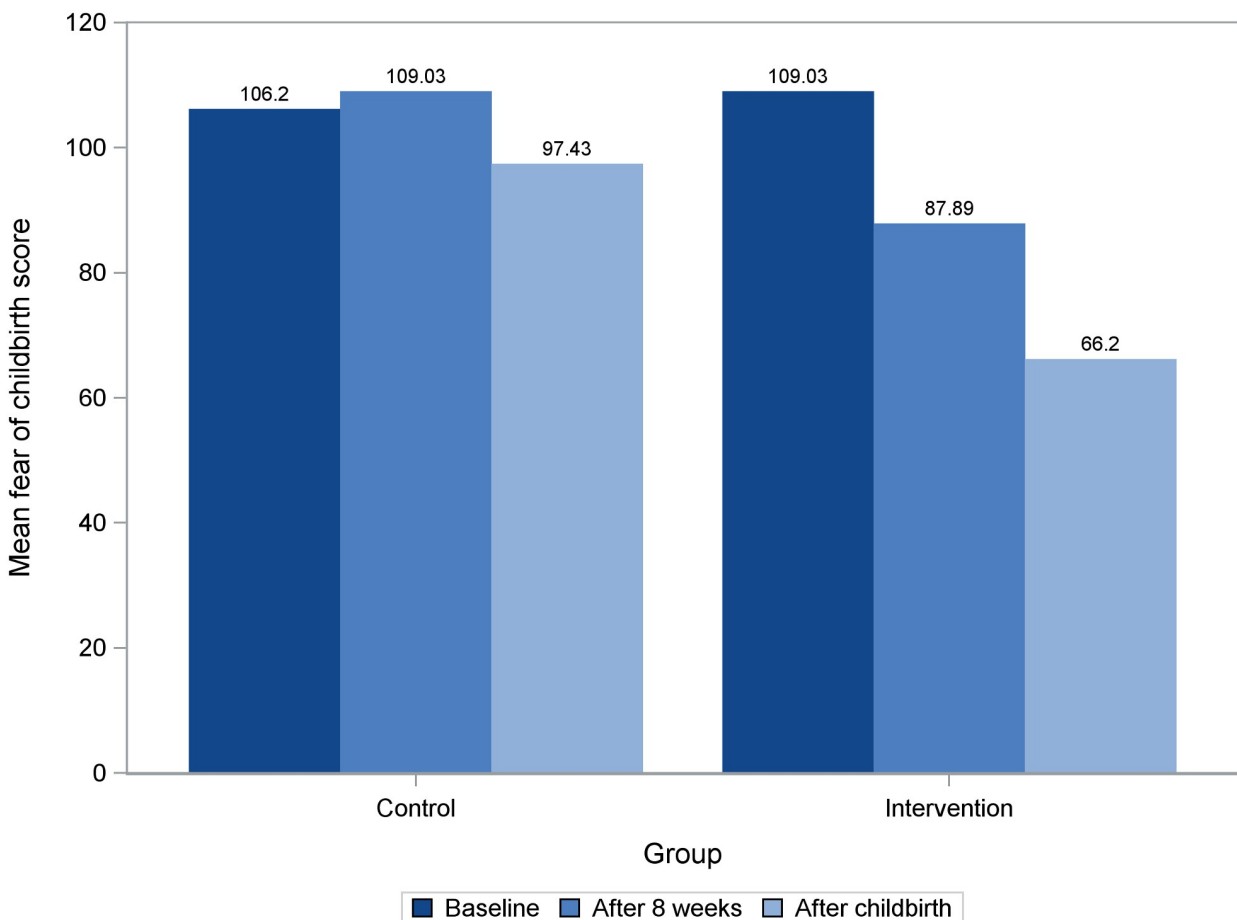

**Fig 2. Comparison of FOC scores in intervention and control groups at baseline, after 8 weeks and after childbirth.**

## Childbirth self-efficacy

There was no statistically significant difference in mean scores of childbirth self-efficacy between the intervention and control groups at the baseline (p = 0.07). According to the result of linear regression it is estimated that on average those in the intervention group have an 85.60scores more than those in the control group after 8 weeks of intervention (95% CI [80.56, 90.64], P value <0.0001, $R^2$ = 0.94) (Table 4). The results remained constant after adjusting for age, income, education, and baseline childbirth self-efficacy score (S3 Table in S3 File). The comparison of the mean score of childbirth self-efficacy and its subscales in the intervention and control groups are reported in S4 Table in S3 File.

## Birth mode

Two intervention and control groups had a statistically significant difference in their birth mode (p = 0.03). The intervention group had a lower rate of CS than the control group (31.4% in the intervention group versus 60% in the control group). The result of the logistic regression showed that the odds of CS are approximately 3.27 times higher in the control group compared to the intervention group (95% CI 1.22, 8.75). The adjusted model for age, family income, education showed similar results.

## Discussion

This study aimed to determine the effect of an interactive mHealth application named Tele-midwifery based on education and continuous support provided by midwives on FOC, childbirth self-efficacy, and birth mode in primiparous women. Based on the results of this study, Tele-midwifery application reduced FOC, increased childbirth self-efficacy, and decreased the CS rate among the intervention group compared to the control group.

Women in the intervention group received educational materials, including a broad range of information regarding pregnancy and childbirth, in order to enhance their knowledge and eliminate their misconceptions about childbirth, which contribute to FOC [34]. Education will assist women in addressing their concerns regarding childbirth, as well as providing them with confidence that their children and themselves will not be in danger. This will help them in coping with their fears related to childbirth [35]. Our findings are in line with previous studies which used educational content as their intervention for reducing FOC, increasing self-efficacy, and decreasing the number of CS [26, 36, 37]. The results of the Shahsavan et al. (2021) study also showed that an educational program based on cognitive-behavioral therapy significantly reduced the FOC in pregnant women [38]. However, the reduction in FOC score in Shahsavan et al. study was less than in our study, which can be due to the lower baseline FOC score and the difference in the educational training in Shahsavan et al. study compared to ours. The prominent role of continuous support and online services in our study compared to the Shahsavan

**Table 4. Effect of mHealth application on self-efficacy score in intervention group compared to control group at baseline and 8 weeks after intervention (n = 70).**

| Time/Group | Coefficient | SE | 95% CI | | P-Value |
|---|---|---|---|---|---|
| | | | LL | LL | |
| **Baseline** | | | | | |
| Intervention vs. Control | 3.60 | 1.98 | -0.35 | 7.55 | 0.07 |
| **After 8 weeks** | | | | | |
| Intervention vs. Control | 85.60 | 2.55 | 80.56 | 90.64 | <0.0001 |

SE = Standard Error, CI = Confidence Interval, LL = lower Limit, UL = Upper Limit

et al. study can be another reason for this discrepancy. Toohl et al. (2014) used a telephone-based psycho-education intervention offered by midwives to reduce FOC and increase self-efficacy [37]. The intervention used in their study is similar to the present study in terms of providing virtual education and counseling by midwives. However, the mean score of increased childbirth self-efficacy in Toohl et al. was less than in our study. This difference can be due to the shorter intervention duration (only two telephone sessions in Toohl et al. study) and lack of continuous support compared to the present study. FOC is negatively associated with childbirth self-efficacy and CS in pregnant women. The reduction in FOC may have had a significant effect on improving women's self-efficacy and reducing the number of CS in the intervention group compared to the control group. Fearful pregnant women consider childbirth as a complicated and challenging process and beyond their abilities, which contributes to a reduced childbirth self-efficacy, and higher number of CS [39]. According to Taheri Z et al. (2014) [40] and Stoll k et al. (2015) [41] the lower FOC and increased self-efficacy were associated with a lower CS rate among pregnant women. These results were consistent with the present study, as the CS rate in women in the intervention group was 28.6% higher than the control group.

Our intervention's emphasis on continuity of care may be another explanation for the reduced FOC in the intervention group compared with the control group at the end of pregnancy and after childbirth [42]. Consistent with the present study, Hildingsson et al. (2018) used the case-load midwifery method and continuous support during pregnancy and childbirth, as well as counseling women to eliminate the causes of FOC, which reduced the fear in women [43].

The method used in this study was a mHealth-based intervention that provided remote access to evidence-based healthcare, such as education and consultation for women with FOC. Using the mobile health applications to provide healthcare is an effective means of ensuring that professional healthcare is available, affordable, and accessible to all individuals, particularly pregnant women [44]. The intervention used in this study addressed the problem of limited access to health care, particularly during the COVID-19 pandemic, when in-person prenatal care decreased, without an alternative that could provide remote care [45]. Furthermore, mHealth care reduces commuting and absence time from work for pregnant women [44], in rural areas in particular, where the shortage of health care providers and limited transportation lead to adverse outcomes for both women and children [46]. Multiple studies have shown that mHealth methods are proper for prenatal care, such as education and consultation [47, 48]. Our findings are also consistent with these results. Thus, the method used in this study can be considered a beneficial approach for offering care to pregnant women with FOC.

## Strengths and limitations of research

This study was conducted during the COVID-19 pandemic; thus, it can be complicated to generalize the study's results to normal conditions. Lack of access to childbirth preparation classes and the high risk of getting infected by in-person visits caused pregnant women to raise more concerns about their health. Therefore, one of the possible explanations for adherence to intervention, may be the accessibility of Tele-midwifery intervention during the COVID-19 crisis, which answered their needs and provided continuous care.

In this study we did not capture the frequency and nature of interactions between participants and healthcare providers who delivered the continues support feature of mHealth intervention through the application. Consequently, the specific types of support, the average frequency of provider contact (such as interactions per day or week), and the potential impact of these interactions on maternal health outcomes remain unexplored in this study. Future research is needed to delve deeper into understanding the dynamics and potential benefits of such interactions for expecting mothers.

According to the latest statistics available to the public, in Baharlou Hospital, there are approximately 350 births each month, comprising 38% CS [49]. While compared to the pre-pandemic period, the rate of CS increased during the COVID-19 pandemic [50], the high rate of CS in Iran has always been a challenge in the field of maternal health [51]. Although this study suggests a possible approach to reduce the number of CS, the rate of CS in this study, and overall in Iran [52] is still higher than the world health organization (WHO) recommendation (maximum of 15%) [53], and further research is needed to identify contributing factors to this high number of CS.

## Implications for future research

In this study, participants received continuous support for addressing their questions throughout the week. However, replicating such personalized attention on a larger scale could be challenging. Real-world implementation might lack sufficient qualified healthcare professionals in the field of obstetrics, potentially affecting intervention quality and adherence. Additionally, strategies to bridge the gap between controlled settings and real-world scenarios during scaling up need to be explored to maintain intervention quality. Future studies are also needed to investigate the effects of this study's intervention by considering additional variables such as the number of antenatal care visits and child outcomes, including stillbirth and preterm birth. Recruiting multiparous and primiparous into future research can also be another area before implementing this study intervention in real settings.

## Conclusion

In this randomized clinical trial, women who received the Tele-midwifery intervention reported less FOC and more childbirth self-efficacy after receiving the Tele-midwifery application. Furthermore, the rate of CS decreased in the intervention group. Based on the results, the mHealth application can be considered a supportive treatment for women diagnosed with FOC childbirth, especially during crises such as the COVID-19 pandemic, which limits access to health settings due to preventive measures.

## Supporting information

**S1 File. Protocol of the study.**
(DOCX)

**S2 File. CONSORT checklist.**
(DOC)

**S3 File. S1-S4 Tables.**
(DOCX)

## Acknowledgments

We wish to thank all the participants for generously contributing their time and experience to this study. We are grateful for the support received from all the members of the Obstetrics and Gynecology Department and Bahar clinic in Baharlou hospital.

## Author Contributions

**Conceptualization:** Sahar Khademioore, Elham Ebrahimi, Shohreh Movahedi.

**Data curation:** Sahar Khademioore, Ahmad Khosravi.

**Formal analysis:** Sahar Khademioore, Ahmad Khosravi.

**Investigation:** Sahar Khademioore, Elham Ebrahimi.

**Methodology:** Sahar Khademioore, Elham Ebrahimi, Ahmad Khosravi.

**Project administration:** Sahar Khademioore, Elham Ebrahimi, Shohreh Movahedi.

**Resources:** Sahar Khademioore.

**Software:** Sahar Khademioore.

**Supervision:** Sahar Khademioore, Elham Ebrahimi, Ahmad Khosravi, Shohreh Movahedi.

**Writing – original draft:** Sahar Khademioore, Elham Ebrahimi.

**Writing – review & editing:** Sahar Khademioore, Elham Ebrahimi, Ahmad Khosravi, Shohreh Movahedi.

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
