## [Decision Letter · Decision Letter 0]

25 Jul 2023

PONE-D-23-17998The Effect of an mHealth Application based on Continuous Support and Education on Fear of Childbirth, Self-efficacy, and Birth Mode in Primiparous Women: A Randomized Controlled TrialPLOS ONE

Dear Dr. Khademioore,

Thank you for submitting your manuscript to PLOS ONE. After careful consideration, we feel that it has merit but does not fully meet PLOS ONE’s publication criteria as it currently stands. Therefore, we invite you to submit a revised version of the manuscript that addresses the points raised during the review process.

Please adress each of the reviewers' concerns. 

We look forward to receiving your revised manuscript.

Kind regards,

Hector Lamadrid-Figueroa, MD, ScD

Academic Editor

PLOS ONE

Additional Editor Comments:

Thank you for giving us the opportunity to consider your work. Please address the reviewers' concerns.

Reviewers' comments:

Reviewer's Responses to Questions

**Comments to the Author**

1. Is the manuscript technically sound, and do the data support the conclusions?

Reviewer #1: Yes

Reviewer #2: Yes

Reviewer #3: Partly

2. Has the statistical analysis been performed appropriately and rigorously? 

Reviewer #1: I Don't Know

Reviewer #2: Yes

Reviewer #3: No

3. Have the authors made all data underlying the findings in their manuscript fully available?

Reviewer #1: No

Reviewer #2: Yes

Reviewer #3: Yes

4. Is the manuscript presented in an intelligible fashion and written in standard English?

Reviewer #1: Yes

Reviewer #2: Yes

Reviewer #3: Yes

5. Review Comments to the Author

Reviewer #1: Overall, the authors provide a well-conceived, well-presented report of a randomized clinical trial to assess the impact of a mHealth intervention on Fear of Childbirth and other outcomes with important and significant findings. This is an important addition to the literature on interventions that may improve the quality of antenatal care and impact CS rates. There are however, a number of areas that need further clarification or modification.

Background:

First, it would be valuable to have greater contextual information on the setting in which the study took place. For example, at this particular hospital what is:

1. the overall volume of deliveries and cesarean section rate among all women presenting for care and during the period of the study (What was the impact of COVID?).

2. Who usually cares for women during prenatal care, birth and postnatally- midwives, physicians?

3. What is the usual number of ANC visits in general, and more specifically for the participants in the study what number of ANC visits did the intervention and control groups attend (results) ?

Methods:

Why was the intervention limited to women 18-40?

Line 99 …..the authors write with regard to inclusion criteria…..”as well as the ability to work with them, having access to the internet, not having chronic diseases and CS indications before or during pregnancy. The exclusion criteria were, emergency pregnancy conditions that required intervention (e.g., severe symptoms of COVID-19, placental abruption, fetus abnormalities, preeclampsia).” Did any of this occur after enrollment? What happened if these developed during the course of the study?

Line 202 The authors state that the researchers were available every day 7 am to 9 pm to answer questions posed in the intervention group. It appears that only 2 of the researchers are clinicians, did they respond to all of the questions? 7 days a week?

Sample size. The authors fail to describe how the data from the 2016 study by Gozde informed the sample size calculation- a basis for underlying prevalence of FOC?

Results:

The authors do not present the number of ANC visits attended in each group. Was there any difference in ANC attendance and may that have been a factor in outcomes? If overall there were very few in person visits in both groups due to COVID, then this study does not actually assess impact of the intervention vs standard care it is intervention vs decreased care.

Process information. Granted, there may be a secondary manuscript in the works that describes the actual dose of the intervention. However, the access to providers via the application to answer questions and respond to concerns is mentioned but not quantified. It would be useful to know how important this feature may be and how labor intensive. It would be valuable to have a sense of how frequently and for what reasons providers were contacted. Were there any referrals made to emergency services through this? Similarly, there is no data provided regarding the frequency with which women read the messages in a timely fashion and accessed the educational modules.

It would be valuable to have an overall sense of birth outcomes among the participants. Live births, stillbirths, early neonatal deaths, preterm birth

The authors maintained commendable follow up with no losses. I wonder what happened to women who developed pregnancy complications after enrollment- I presume they were kept in the intervention arm? Was this comparable in the groups?

Discussion.

Is there any information on the indications for CS? The authors rightly point out that the underlying CS rate is far above that considered appropriate for primiparous women. How might this have impacted the success of the intervention. Are there other interventions or movement to address the high CS rate in this setting? How might, or did COVID, impact this?

I am curious about the impact of the access to providers for questions and concerns. Is it possible to tease out how important the access compared to the educational components is?

The authors to not share their opinions on how this study should inform future work or if there are plans to study this intervention at scale in more of a real world setting. Granted this worked in a highly controlled setting where researchers were available to respond to questions but it is unclear what would happen in a scaled intervention setting.

Data:

The authors do not make the data available.

Reviewer #2: This manuscript reports the results investigating the effect of an mHealth application based on continuous support and education on primiparous women who were fear of childbirth on self-efficacy and birth mode with a randomized controlled study. I have below minor comments.

Please make a clear statement about which measurement or outcome was used for sample size estimate. How to come with the difference of 21 and sd of 7.2 for that outcome? Pleas note, since the difference of 21 is very big but the variation is small, the n=35/group is far greater than a sample size that achieves 90% power at alpha=0.05.

Table 2, the unit of gestational age should be weeks.

Line 253, should “increase” be “decrease”?

Line 265, Should “Two” be “The”?

Reviewer #3: Thank you for an interesting paper to review. I think it is a well-written paper with a novel and important approach for pregnant families. Nevertheless, the analysis seems to be relatively simple. I highly recommend using a logistic regression model as a minimum and then presenting your new results.

6. PLOS authors have the option to publish the peer review history of their article (what does this mean?). If published, this will include your full peer review and any attached files.

Reviewer #1: No

Reviewer #2: No

Reviewer #3: No

---

## [Author Response · Author response to Decision Letter 0]

9 Aug 2023

Reviewers' comments:

Reviewer #1: 

"Overall, the authors provide a well-conceived, well-presented report of a randomized clinical trial to assess the impact of a mHealth intervention on Fear of Childbirth and other outcomes with important and significant findings. This is an important addition to the literature on interventions that may improve the quality of antenatal care and impact CS rates. There are however, a number of areas that need further clarification or modification.

Background:

First, it would be valuable to have greater contextual information on the setting in which the study took place. For example, at this particular hospital what is:

1. the overall volume of deliveries and cesarean section rate among all women presenting for care and during the period of the study (What was the impact of COVID?).

2. Who usually cares for women during prenatal care, birth and postnatally- midwives, physicians? 

3. What is the usual number of ANC visits in general, and more specifically for the participants in the study what number of ANC visits did the intervention and control groups attend (results) ?"

Authors' Response: Thank you for your valuable feedback and the thorough review of our manuscript. We highly appreciate your insightful comments.

We acknowledge the importance of providing more conceptual information on the study setting. However, as the other related information is mentioned in the Methods and Discussion section, we also added points raised in numbers 1-3 in a related section to enhance the conciseness and clarity of the manuscript, and we hope these changes are satisfactory. The related sections now read as follows:

"According to the latest statistics available to the public, in Baharlou Hospital, there are approximately 350 births each month, comprising 38% CS (1). Although compared to the pre-pandemic period, the rate of CS increased during the COVID-19 pandemic (2), the high rate of CS in Iran has always been a challenge in the field of maternal health."

"The routine prenatal care in Iran includes up to 8 prenatal visits, which in the study site would be provided to all pregnant individuals by midwives and obstetricians. Routine prenatal care includes regular visits to monitor women and fetal health."

We also explained more about the number of antenatal care (ANC) visits in the study in the following comments.

"Methods:

Why was the intervention limited to women 18-40?"

Authors' Response: Our study's primary focus was on low-risk pregnancies. As a result, we excluded adolescent pregnancies (under 18 years old) and pregnancies in older age groups that are at higher risk of pregnancy complications and need medical interventions. Based on the result of this study, the age range of participants included in the analysis was limited to 18-33 years. This age range is considered low risk for pregnancies, and we attribute this to our study's emphasis on primiparous women, who typically have a lower mean age than multiparous women.

"Line 99 …..the authors write with regard to inclusion criteria….." as well as the ability to work with them, having access to the internet, not having chronic diseases and CS indications before or during pregnancy. The exclusion criteria were, emergency pregnancy conditions that required intervention (e.g., severe symptoms of COVID-19, placental abruption, fetus abnormalities, preeclampsia)." Did any of this occur after enrollment? What happened if these developed during the course of the study?"

Authors' Response: We specified these exclusion criteria in our protocol, and participants meeting these conditions were to be excluded from the study. The rationale behind these exclusions was that these conditions involve severe complications requiring emergency care, making them unsuitable for our intervention. Additionally, such conditions may lead to heightened anxiety and fear among individuals.

However, none of these conditions occurred among our participants. One possible reason for this is our eligibility criteria, which excluded high-risk pregnancies. In addition, many of these conditions could have arisen before the 26th week of gestation, when the eligibility criteria were assessed, resulting in the non-recruitment of participants who developed these conditions for the study. We have also documented these details in our CONSORT flow diagram (Figure 1).

"Line 202 The authors state that the researchers were available every day 7 am to 9 pm to answer questions posed in the intervention group. It appears that only 2 of the researchers are clinicians, did they respond to all of the questions? 7 days a week?"

Authors' Response: Except for our statistician (AK), the other three authors (SK, EE, and SM) are a certified midwife, a doctor of reproductive health (MSc in Midwifery), and an obstetrician and gynecologist. They are fully capable and eligible to address participants' pregnancy-related questions.

One of the most essential concepts of our intervention is continuous care, which is why the researchers offered uninterrupted support throughout the week, and the researchers answered any questions that might have been raised based on the educational content or general questions regarding their pregnancies, such as the number of in-person visits needed or specific types of food that should be avoided, etc. 

"Sample size. The authors fail to describe how the data from the 2016 study by Gozde informed the sample size calculation- a basis for underlying prevalence of FOC?"

Authors' Response: Thank you for bringing this to our attention. We appreciate your comment and acknowledge that it is necessary to describe how the cited paper was used to estimate the sample size, outcome, and scale used. 

The mean difference in fear of childbirth on the Wijma Delivery Expectancy/Experience Questionnaire (W-DEQ) in the intervention and control group in the cited paper was used to estimate the sample size. We have updated the manuscript and included this information.

"Results:

The authors do not present the number of ANC visits attended in each group. Was there any difference in ANC attendance and may that have been a factor in outcomes? If overall there were very few in person visits in both groups due to COVID, then this study does not actually assess impact of the intervention vs standard care it is intervention vs decreased care."

Authors' Response: This is a very interesting point that you brought up. However, capturing the number of ANC visits was beyond the scope of this study. In addition, we want to clarify that our intervention was not intended to substitute in-person ANC visits; it was completely independent of ANC visits and served a different purpose. While we provided continuous care and addressed participants' pregnancy-related questions, we never offered formal ANC procedures such as prescribing medication, requesting screenings, sonograms, or lab tests. As a result, we assume both groups had a similar chance of missing their ANC visits due to any reason and, more specifically, because of the COVID-19 pandemic. 

Additionally, it is essential to consider the study timeline. Our research was conducted from February to April. During almost half of the study, COVID-19 had not completely taken over Iran and affected routine care. However, we confirm that antenatal care was continuously offered in the prenatal clinic of the study site, even during the pandemic.

"Process information. Granted, there may be a secondary manuscript in the works that describes the actual dose of the intervention. However, the access to providers via the application to answer questions and respond to concerns is mentioned but not quantified. It would be useful to know how important this feature may be and how labor intensive. It would be valuable to have a sense of how frequently and for what reasons providers were contacted. Were there any referrals made to emergency services through this? Similarly, there is no data provided regarding the frequency with which women read the messages in a timely fashion and accessed the educational modules."

Authors' Response: Thank you for your attention to detail. In this current manuscript, we have presented all the data we collected on this intervention. 

Considering the multifactorial pathway of fear of childbirth, our intervention included various components to address the diverse factors contributing to fear in pregnant individuals. During the study's design stage, we aimed for a complex intervention, believing that the successful reduction in fear of childbirth resulted from all the intervention components working together. At this stage of testing this new intervention, our primary goal was not to examine the isolated effects of each component, including education and continuous support provided by a care provider to answer questions.

It is important to clarify that this randomized controlled trial was conducted as part of a master's degree thesis with limited budget and time constraints, which unfortunately did not allow us to expand our work and collect more data. However, we totally agree on the importance of collecting more data on each aspect of the intervention and now that the results of this work have shown the possibility of success in reducing fear of childbirth, further comparative research are needed to build upon these findings to determine the most successful component, labor intensiveness (cost-effectiveness) and even explore women's perspectives through qualitative research before implementing the intervention in real-world settings. As previously mentioned, our intervention was not intended as an alternative to antenatal care, but rather as an addition to it. Similar to any educational content our intervention also needed a feature to answer questions raised by our educational material.

Furthermore, although we were prepared to refer participants in case of an emergency, no such contacts were made during the study period. We added this information to the manuscript. 

Lastly, in our manuscript, we have addressed your last question: 

"The application's educational content was designed for eight weeks in a way that between 3-4 short messages were sent to women on a daily basis." and "The researcher could track the women's activities in the application, and in case the women had not use the application for more than two days, a reminder was sent to follow them up." 

"It would be valuable to have an overall sense of birth outcomes among the participants. Live births, stillbirths, early neonatal deaths, preterm birth"

Authors' Response: It would indeed be valuable information and an addition to the literature, making it a potential focus for future studies. However, it is important to note that the child outcome was not the objective of this study. Our main emphasis was on maternal outcomes, specifically fear of childbirth, self-efficacy, and childbirth mode. Child outcomes were not listed from the beginning in our ethics approval and protocol, and as a result, were not captured in our data.

"The authors maintained commendable follow up with no losses. I wonder what happened to women who developed pregnancy complications after enrollment- I presume they were kept in the intervention arm? Was this comparable in the groups?"

Authors' Response: As mentioned in the first paragraph of the Results section and illustrated in our CONSORT diagram, we did not experience any loss to follow-up or exclusion due to severe pregnancy complications requiring further medical attention and all participants who were initially enrolled were included in the final analysis. Our reasoning behind this decision was explained earlier in response to your previous question regarding exclusion criteria. 

"Discussion.

Is there any information on the indications for CS? The authors rightly point out that the underlying CS rate is far above that considered appropriate for primiparous women. How might this have impacted the success of the intervention. Are there other interventions or movement to address the high CS rate in this setting? How might, or did COVID, impact this?"

Authors' Response: Thank you for raising this interesting point. The C-section rate in Iran has seen a significant increase over the years, rising from 35% in 2005 to 48% in 2014 and even more in private settings (3), which is considerably high compared to the WHO recommendation. This high rate of C-sections is a critical healthcare issue in the system and has been the subject of investigation and interventions to address this concern. For example, in governmental hospitals, elective C-sections are not possible based on maternal request. However, at the time of our study, no other intervention was ongoing with the purpose of reducing C-sections at the study site. It is important to note that the high rate of C-sections is not specific to our study setting; rather, it is prevalent in many hospitals. Considering this context, we view our intervention as successful in reducing the number of C-sections in the intervention group compared to the control group. 

As our study is a governmental hospital, the indication for primary C-sections is fetal distress or maternal emergencies. However, recording the specific indication of C-sections study was beyond the scope of our study.

Regarding the impact of COVID-19 on the C-section rate, it is essential to clarify that none of our participants suffered from COVID-19 infection, and they would have been excluded if they had. Therefore, based on the results of this study, we cannot draw conclusions about the specific impact of COVID-19 on the C-section rate.

Nonetheless, we agree that this could be an interesting and multifactorial aspect to investigate in future research, and we have included it as an implication for future studies in our manuscript.

"I am curious about the impact of the access to providers for questions and concerns. Is it possible to tease out how important the access compared to the educational components is?"

Authors' Response: As we comprehensively explained above, we believe our intervention is effective as a composite of different components and did not capture the effect of each component separately. However, we strongly suggest future studies take this into consideration and have added this concern to our implications section in the manuscript.

"The authors to not share their opinions on how this study should inform future work or if there are plans to study this intervention at scale in more of a real world setting. Granted this worked in a highly controlled setting where researchers were available to respond to questions but it is unclear what would happen in a scaled intervention setting."

Authors' Response: Based on the findings of our study and your valuable comments regarding other variables that are better to consider when investigating the effect of this intervention, such as access to ANC, child outcome, and each component of the intervention, we added a section under the implications for future research to inform researchers.

We also provided information for discussing the limitations and considerations for scaling up the intervention, including how the availability of researchers in the controlled setting might differ from a real-world scenario.

"Implications for future research

In this study, participants received continuous support for addressing their queries throughout the week. However, replicating such personalized attention on a larger scale could be challenging. Real-world implementation might lack sufficient research personnel, potentially affecting intervention quality and adherence. Future research should investigate each component of our study intervention. Additionally, strategies to bridge the gap between controlled settings and real-world scenarios during scaling up need to be explored to maintain intervention quality.

Future studies are also needed to investigate the effects of this study's intervention by considering additional variables such as the number of antenatal care visits and child outcomes, including stillbirth and preterm birth. Recruiting multiparous and primiparous into future research can also be another area before implementing this study intervention in real settings."

"Data:

The authors do not make the data available."

Authors' Response: We confirm that all the data collected in this study has been thoroughly presented within the manuscript, including the text, tables, and figures.

However, as the study was conducted in a single center and within a relatively short timeframe, there is a potential risk of participant identification if the data set will make publicly available. Considering the PLOS ONE policy, "For studies involving human research participant data or other sensitive data, we encourage authors to share de-identified or anonymized data. However, when data cannot be publicly shared, we allow authors to make their data sets available upon request", due to the restrictions regarding participant identification, the data set will be available upon request from the corresponding author. 

Reviewer #2: 

"This manuscript reports the results investigating the effect of an mHealth application based on continuous support and education on primiparous women who were fear of childbirth on self-efficacy and birth mode with a randomized controlled study. I have below minor comments.

Please make a clear statement about which measurement or outcome was used for sample size estimate. How to come with the difference of 21 and sd of 7.2 for that outcome? Pleas note, since the difference of 21 is very big but the variation is small, the n=35/group is far greater than a sample size that achieves 90% power at alpha=0.05."

Authors' Response: We appreciate your attention to detail and input regarding the sample size estimation. The cited paper used an intervention similar to our study's intervention; therefore, we used their study as a basis for calculating the minimum fear of childbirth score difference between the intervention and control groups. According to Gözde Birsbir et al.'s (2016), the difference in FOC between groups post-intervention was 21 scores (m i = 25.5, m c= 46.5). 

With regard to your second point, we appreciate that you captured this point. The SD mentioned in the manuscript was the differences of the SD between the two groups based on the study used for sample size calculation (s i=18.2 , s c=25.4). We have revised the manuscript to eliminate any confusion and provide a clearer statement regarding the sample size calculation.

Also, our sample size calculation is noted below:

Given the:

α=0.05 

β=0.9 

μ1=25.5 

μ2=46.5 

s1=18.2 

s2=25.4

We calculate the sample size using the below formula. 

n=(〖( Z_(1-α⁄2)+ Z_(1-β))〗^2.(S_1^2+S_2^2))/(μ_1- μ_1 )^2 

n=(〖( 1.96 - (-1.28)〗^2.(〖18.2〗^2+〖25.4〗^2))/(25.5- 46.5)^2 

Therefore, we estimated to have 35 participants in each group.

"Table 2, the unit of gestational age should be weeks."

Authors' Response: Thank you for capturing this. The unit has been corrected.

Line 253, should "increase" be "decrease"? 

Authors' Response: The correct word is "increase." This accurately reflects that the gap between the intervention and control groups in terms of fear of childbirth scores widened as time progressed, indicating a stronger and more significant effect of the intervention. We appreciate your keen observation, and to avoid confusion, we have added a summary to emphasize this aspect. We also added a figure to visualize the change in FOC in the intervention and control groups in different time points of the study.

"Line 265, Should "Two" be "The"?"

Authors' Response: Thank you again for your input, and we have corrected the title.

Reviewer #3: 

"Thank you for an interesting paper to review. I think it is a well-written paper with a novel and important approach for pregnant families. Nevertheless, the analysis seems to be relatively simple. I highly recommend using a logistic regression model as a minimum and then presenting your new results."

Authors' Response: Thank you sincerely for taking the time to review our manuscript.

We acknowledge your concern regarding our analysis. For our secondary outcome, we agree that we can add more analysis and have included a logistic regression to estimate the odds of CS in the intervention group compared to the control group. The results have been added to the manuscript. Regarding the primary outcome, FOC, we opted for a generalized linear model (repeated-measures ANOVA) due to the longitudinal nature of our data. This approach allowed us to analyze the outcome (FOC) measured over time on the same individuals at different time points (baseline, after 8 weeks, and after childbirth). The repeated-measures ANOVA effectively accounts for the correlation between these data points.

Furthermore, it is worth noting that ANOVA can indeed be seen as a specific case of regression analysis. In our study, we chose the repeated-measures ANOVA method because it offered us the ability to handle within-subject variability and gain valuable insights into the changes over time. By utilizing repeated-measures ANOVA, we were able to capture the temporal evolution of FOC in both the intervention and control groups.

While adding a regression analysis might yield similar results, we believe that the repeated-measures ANOVA was the most appropriate choice for our study's objectives and data structure. It allowed us to examine the effects of time and group on FOC, while also accounting for the repeated measurements within the same individuals.

We hope this explanation clarifies our analysis rationale, and we are open to any further suggestions or inquiries you may have.

References

1. The statistics of patients at Labor and the increase in natural childbirth at Baharlou Hospital. 2009 [Available from: http://194.225.51.22/baharlou].

2. Gharacheh M, Kalan ME, Khalili N, Ranjbar F. An increase in cesarean section rate during the first wave of COVID-19 pandemic in Iran. BMC Public Health. 2023;23(1):936.

3. Shahshahan Z, Heshmati B, Akbari M, Sabet F. Caesarean section in Iran. The Lancet. 2016;388(10039):29-30.

---

## [Decision Letter · Decision Letter 1]

29 Aug 2023

PONE-D-23-17998R1The Effect of an mHealth Application based on Continuous Support and Education on Fear of Childbirth, Self-efficacy, and Birth Mode in Primiparous Women: A Randomized Controlled TrialPLOS ONE

Dear Dr. Khademioore,

Thank you for submitting your manuscript to PLOS ONE. After careful consideration, we feel that it has merit but does not fully meet PLOS ONE’s publication criteria as it currently stands. Therefore, we invite you to submit a revised version of the manuscript that addresses the points raised during the review process.

We look forward to receiving your revised manuscript.

Kind regards,

Hector Lamadrid-Figueroa, MD, ScD

Academic Editor

PLOS ONE

Journal Requirements:

Additional Editor Comments:

Please address outstanding issues raised by reviewers 1 and 3.

Reviewers' comments:

Reviewer's Responses to Questions

**Comments to the Author**

1. If the authors have adequately addressed your comments raised in a previous round of review and you feel that this manuscript is now acceptable for publication, you may indicate that here to bypass the “Comments to the Author” section, enter your conflict of interest statement in the “Confidential to Editor” section, and submit your "Accept" recommendation.

Reviewer #1: (No Response)

Reviewer #2: All comments have been addressed

Reviewer #3: All comments have been addressed

2. Is the manuscript technically sound, and do the data support the conclusions?

Reviewer #1: Yes

Reviewer #2: (No Response)

Reviewer #3: Yes

3. Has the statistical analysis been performed appropriately and rigorously? 

Reviewer #1: I Don't Know

Reviewer #2: (No Response)

Reviewer #3: No

4. Have the authors made all data underlying the findings in their manuscript fully available?

Reviewer #1: Yes

Reviewer #2: (No Response)

Reviewer #3: No

5. Is the manuscript presented in an intelligible fashion and written in standard English?

Reviewer #1: Yes

Reviewer #2: (No Response)

Reviewer #3: Yes

6. Review Comments to the Author

Reviewer #1: The authors have done a good job in responding to the majority of my prior comments and I appreciate their efforts.

There remain however, a couple of outstanding questions and suggestions.

1. Methods. It is important that the methods are described in such a way that the intervention could be replicated. After learning that this is a Master’s project, I am left wondering who and how the educational resources were created?- this is a very labor intensive effort. Are the videos, messages, and images all original content? I would like to know how many videos, images, texts were created or included? This can be quantified and added to table 1. If the program linked to any pre-existing videos or materials, they should be referenced.

2. In the Methods section titled, “Continuous support and engagement of women:”, where the authors state (pg 11 line 205) “to provide women with an accessible source of information about their pregnancy and childbirth decisions"…The authors could clarify that the support did not include any prescription of medications or referrals for labs or testing of any kind.

3. I understand that the project did not have the resources to analyze frequency and type of contacts (dose and content of support messaging) with healthcare providers. However, I think this is an important issue for discussion in the limitations/future research section to add a statement acknowledging that frequency and content of support from obstetric providers and use of messaging were not measured. If the authors are able to add any assessment in the limitations section about how frequently they were contacted- i.e. -on average providers estimated they were contacted 1-5 times per day, 20 times a day, once a week…..anything would help.

4. On pg 20 line 349- Future research -need to consider not availability of “research personnel” but rather availability of qualified obstetric providers who can provide the education and advice needed.

Reviewer #2: (No Response)

Reviewer #3: Thank you for addressing all of our comments. However, I believe that your statistical analysis could be more robust by utilizing logistic regression models instead of ANOVA. Your statistical advisor should be likely aware that a logistic regression can be employed for any longitudinal study to measure the effects while controlling for potential confounders. An alternative model you could use to adjust is a Poisson model. Although, you have made an effort to include one logistic regression model in your analysis. Just make sure to incorporate a table presenting these results. This will ensure clarity on the variables you've included for adjustment. If the limits on tables/word count prevent this, consider at least mentioning in your results section which variables you used for adjustment and add this table in the Supporting information section.

In line 227: Please consider to add the words "A logistic regression MODEL was also PERFORMED to estimate the odds of CS in the intervention group..." to complete your sentence.

7. PLOS authors have the option to publish the peer review history of their article (what does this mean?). If published, this will include your full peer review and any attached files.

Reviewer #1: No

Reviewer #2: No

Reviewer #3: No

---

## [Author Response · Author response to Decision Letter 1]

21 Sep 2023

Reviewer #1: 

“Reviewer #1: The authors have done a good job in responding to the majority of my prior comments and I appreciate their efforts. There remain however, a couple of outstanding questions and suggestions.”

“1. Methods. It is important that the methods are described in such a way that the intervention could be replicated. After learning that this is a Master’s project, I am left wondering who and how the educational resources were created? - this is a very labor intensive effort. Are the videos, messages, and images all original content? I would like to know how many videos, images, texts were created or included? This can be quantified and added to table 1. If the program linked to any pre-existing videos or materials, they should be referenced.”

Authors' Response: Thank you for your comment. Over the past years, Iran has witnessed a range of initiatives aimed at producing health-related materials across various fields. The first author of this paper (SK) has actively participated in such endeavors, particularly in the field of midwifery, with the objective of providing her patients with the most up-to-date, evidence-informed recommendations. Additionally, although creating content for a mHealth application can be labor-intensive, these materials can be used many times, making mHealth applications more cost-effective in the long term. In this project, the research team meticulously curated and organized existing materials while also incorporating fresh information from diverse scientific sources, spanning from textbooks to comprehensive Cochrane systematic reviews. We have included the list of references used in the method section. It's important to note that we refrained from utilizing any pre-existing materials to uphold copyright integrity. The number of videos, images, and text messages has been added to Table 1. 

“2. In the Methods section titled, “Continuous support and engagement of women:”, where the authors state (pg 11 line 205) “to provide women with an accessible source of information about their pregnancy and childbirth decisions"…The authors could clarify that the support did not include any prescription of medications or referrals for labs or testing of any kind.”

Authors' Response: Thanks for keen your comment. We have provided this important information to our “Continuous support and engagement of women” section. 

Now it reads:

“However, this feature was not intended to substitute the usual antenatal care, and no medication prescriptions, requesting laboratory tests, or imaging were offered via our mHealth.”

“3. I understand that the project did not have the resources to analyze frequency and type of contacts (dose and content of support messaging) with healthcare providers. However, I think this is an important issue for discussion in the limitations/future research section to add a statement acknowledging that frequency and content of support from obstetric providers and use of messaging were not measured. If the authors are able to add any assessment in the limitations section about how frequently they were contacted- i.e. -on average providers estimated they were contacted 1-5 times per day, 20 times a day, once a week…..anything would help.”

Authors' Response: We have indeed recognized the importance of measuring the frequency and content of support received by participants during the intervention. We have added a paragraph in the limitations section to acknowledge the significance of this aspect. Since we did not collect precise data on the frequency of interactions with healthcare providers via the application, we refrained from providing specific statistics regarding the frequency of contact but instead emphasized its importance for future research. This key point has been highlighted in our discussion. We appreciate your insightful feedback, which has contributed to a more comprehensive understanding of the study's limitations and future research directions.

This section now reads:

“In this study we did not capture the frequency and nature of interactions between participants and healthcare providers who delivered the continues support feature of mHealth intervention. Consequently, the specific types of support, the average frequency of provider contact (such as interactions per day or week), and the potential impact of these interactions on maternal health outcomes remain unexplored in this study. Future research is needed to delve deeper into understanding the dynamics and potential benefits of such interactions for expecting mothers.”

4. On pg 20 line 349- Future research -need to consider not availability of “research personnel” but rather availability of qualified obstetric providers who can provide the education and advice needed.

Authors' Response: Thank you for capturing this. We now used the term 'sufficient qualified healthcare professionals in the field of obstetrics' to better convey our message.

Reviewer #3

“Reviewer #3: Thank you for addressing all of our comments. However, I believe that your statistical analysis could be more robust by utilizing logistic regression models instead of ANOVA. Your statistical advisor should be likely aware that a logistic regression can be employed for any longitudinal study to measure the effects while controlling for potential confounders. An alternative model you could use to adjust is a Poisson model. Although, you have made an effort to include one logistic regression model in your analysis. Just make sure to incorporate a table presenting these results. This will ensure clarity on the variables you've included for adjustment. If the limits on tables/word count prevent this, consider at least mentioning in your results section which variables you used for adjustment and add this table in the Supporting information section.”

Authors' Response: Thank you for your meticulous feedback on our analysis and results section. We have significantly revised the results section to incorporate your suggestions. Now we have presented the result of regression models for all of our outcomes.

To elaborate on our rationale for employing a linear regression model in our analyses, as opposed to logistic regression, we considered the continuous nature of the outcome variables (fear of childbirth and self-efficacy score). Given that our study's primary objective was to capture the degree of change in fear of childbirth and self-efficacy scores, we opted not to categorize our outcome or investigate the intervention's effect on change between categories. This approach allowed us to better account for the variability of outcomes within groups. Additionally, there is no established threshold to categorize self-efficacy from a continuous measure into a categorical variable [1]. Previous research has also treated fear of childbirth and self-efficacy as continuous variables, ensuring improved comparability of our results with existing literature.

Regarding adjustments in our model, the randomization in our study design was designed to balance known and unknown confounders between the two groups. Furthermore, our descriptive analysis in Table 1 revealed no statistically significant differences in known confounders such as age, education, and income levels, and the minor variations between groups were not a cause for concern. In addition, baseline scores for fear of childbirth and self-efficacy were balanced. This suggests the success of our randomization process. Therefore, initially, we presented unadjusted models in our results, as these confounders were anticipated to have a negligible impact on the outcomes [2]. However, in line with your suggestion, we conducted adjusted models accounting for age, income, education, and baseline outcome scores, and we have included this information in the supplementary material. Additionally, in the results section, we have clarified which variables were used for adjusted models, highlighting that the results are presented with these variables held constant.

1- Naggara, O., Raymond, J., Guilbert, F., Roy, D., Weill, A., & Altman, D. G. (2011). Analysis by categorizing or dichotomizing continuous variables is inadvisable: an example from the natural history of unruptured aneurysms. AJNR. American journal of neuroradiology, 32(3), 437–440. https://doi.org/10.3174/ajnr.A2425

2- Vickers A J, Altman D G. Analysing controlled trials with baseline and follow up measurements BMJ 2001; 323 :1123 doi:10.1136/bmj.323.7321.1123

“In line 227: Please consider to add the words "A logistic regression MODEL was also PERFORMED to estimate the odds of CS in the intervention group..." to complete your sentence.”

Authors' Response: Thank you for capturing this. We have revised it accordingly.

---

## [Editor Report · Decision Letter 2]

20 Oct 2023

The Effect of an mHealth Application based on Continuous Support and Education on Fear of Childbirth, Self-efficacy, and Birth Mode in Primiparous Women: A Randomized Controlled Trial

PONE-D-23-17998R2

Dear Dr. Khademioore,

We’re pleased to inform you that your manuscript has been judged scientifically suitable for publication and will be formally accepted for publication once it meets all outstanding technical requirements.

Kind regards,

Hector Lamadrid-Figueroa, MD, ScD

Academic Editor

PLOS ONE
---

## [Editor Report · Acceptance letter]

23 Oct 2023

PONE-D-23-17998R2 

The Effect of an mHealth Application based on Continuous Support and Education on Fear of Childbirth, Self-efficacy, and Birth Mode in Primiparous Women: A Randomized Controlled Trial 

Dear Dr. Khademioore:

I'm pleased to inform you that your manuscript has been deemed suitable for publication in PLOS ONE. Congratulations! Your manuscript is now with our production department. 

Kind regards, 

on behalf of

Dr. Hector Lamadrid-Figueroa 

Academic Editor

PLOS ONE